# Detection of Chlorpyrifos Using Bio-Inspired Silver Nanograss

**DOI:** 10.3390/ma15103454

**Published:** 2022-05-11

**Authors:** Hyunjun Park, Joohyung Park, Gyudo Lee, Woong Kim, Jinsung Park

**Affiliations:** 1Department of Biomechatronics Engineering, Sungkyunkwan University, Suwon 16419, Korea; guswns1105@gmail.com (H.P.); parkjoodori@gmail.com (J.P.); 2Department of Biotechnology and Bioinformatics, Korea University, Sejong 30019, Korea; 3Interdisciplinary Graduate Program for Artificial Intelligence Smart Convergence Technology, Korea University, Sejong 30019, Korea; 4Department of Mechanical Engineering, Hanyang University, Seoul 04763, Korea

**Keywords:** biomimicry, Ag nanograss, chlorpyrifos, surface-enhanced Raman spectroscopy

## Abstract

Chlorpyrifos (CPF) is widely used as an organophosphorus insecticide; however, owing to developmental neurotoxicity, genotoxicity, and other adverse effects, it is harmful not only to livestock but also to humans. Therefore, the use of CPF was recently regulated, and its sensitive detection is crucial, as it causes serious toxicity, even in the case of residual pesticides. Because it is hard to detect the chlorpyrifos directly using spectroscopy (especially in SERS) without chemical reagents, we aimed to develop a SERS platform that could detect the chlorpyrifos directly in the water. In this study, we utilized the intrinsic properties of natural lawns that grow randomly and intertwine with each other to have a large surface area to promote photosynthesis. To detect CPF sensitively, we facilitated the rapid fabrication of biomimetic Ag nanograss (Ag-NG) as a surface-enhanced Raman spectroscopy (SERS) substrate using the electrochemical over-deposition method. The efficiency of the SERS method was confirmed through experiments and finite element method (FEM)-based electromagnetic simulations. In addition, the sensitive detection of CPF was enhanced by pretreatment optimization of the application of the SERS technique (limit of detection: 500 nM). The Ag-NG has potential as a SERS platform that could precisely detect organic compounds, as well as various toxic substances.

## 1. Introduction

Biomimicry is the attempt to solve the technical limitations of engineering problems by mimicking the various features of an ecosystem. Various characteristics of an ecosystem are the optimized results for survival found using natural selection. For example, the most familiar example of biomimicry in daily life is Velcro, which was modeled on the seeds of Serrata. Moreover, waterproof paint from the lotus leaf effect, artificial fibers from the spider silk, adhesive surface from the gecko foot surface, and adhesive materials in water from mussels are famous examples of bio-inspired technology [1,2,3,4,5,6,7,8,9]. As a part of biomimetic technology, we previously developed an artificial microfiber that mimicked a spider’s web and nano pine pollen that mimicked pine pollen [10,11]. Through previous works, we recognized that the biomimicked nanostructure has advantages in the SERS sensing platform especially. Because the biomimetic nanostructures are fabricated using mimicking, the porous or hierarchical structures, such as urchin, pine pollen, and zeolite, not only have a large surface area but also numerous nano-gaps. Numerous nano-gaps in the biomimetic nanostructures induce the highly strong plasmonic ‘hot spots’ and have large surface areas owing to their structural features, resulting in ultra-high sensitivity. In this sense, we focused on the structure of a lawn. A lawn has vertically standing upper lobes and horizontally lying lower lobes acting simultaneously for efficient photosynthesis. The vertically standing upper lobes make it possible that the light can reach the horizontal lower lobes. In addition, the lower lobes have a large surface area for photosynthesis due to the horizontally lying structure. These features of a lawn are an evolutionarily optimized result for survival since it has the maximum photosynthesis efficiency per unit area. From the features of a lawn, we mimicked the structures for enhancing the SERS efficiency. The situation of photosynthesis of a lawn is very similar to a situation where a laser irradiates onto a SERS substrate. Just like the photosynthesis efficiency is enhanced in a lawn, we designed a nanostructure to have a high SERS efficiency by mimicking a lawn.

Chlorpyrifos (CPF) is a representative organophosphorus insecticide that acts on the nervous systems of insects by inhibiting the enzymatic activity of acetylcholinesterase. In detail, CPF induces the metabolite chlorpyrifos-oxon that binds to the acetylcholinesterase and inhibits the activity of the enzyme in the synapse [12,13,14,15]. Humans and livestock that are overexposed to it can develop neurological defects and developmental and immune disorders. Various symptoms, such as headache, vomiting, visual disturbance, paralysis, seizures, and suffocation, are apparent with acute exposure of the human body to CPF. Therefore, its sensitive detection is very important and various studies were conducted in this regard, most of which were through electrochemical techniques that utilized acetylcholinesterase [16,17,18,19], DNA [20,21], and molecularly imprinted polymer (MIP) technology. In particular, acetylcholinesterase, which is related to the main toxicity mechanism of CPF, was used as a probe for detecting it with high sensitivity. However, despite the high sensitivity and field applicability of this electrochemical sensor, the sample preparation was complex. Therefore, to overcome this limitation, the colorimetric method [22,23,24] and SERS [22,25,26,27,28,29,30] techniques were adopted. The SERS-based detection techniques did not require a separate probe to capture targets that were easily detected with high sensitivity because the unique characteristics of CPF could be immediately confirmed in the spectrum. Recent SERS-based CPF detection studies used various nanoparticles composed mostly of Ag, Au, or bimetallic compounds [22,25,26,27,28,29,30]. Qin Xu et al. successfully detected 1 µM of CPF residues in water using gold nano-popcorn [31]. Various methods, such as mass spectroscopy, ultraviolet-visible spectroscopy, and chromatography, were applied; however, each had its disadvantages. Most SERS-based methods used nanoparticles composed of Au or Ag, but the coverage area was relatively narrow with inadequate uniformity; Ag-NG can be easily fabricated with excellent uniformity over a large area, and hence, it is a highly reliable sensor.

In this study, we fabricated bio-inspired Ag-NG by mimicking the structure of a lawn. The strategy for the detection of chlorpyrifos in this report is novel in terms of the SERS structure, which mimicked the structure of a lawn. It causes efficient photosynthesis in the lawn, not only on the upper lobes but also on the lower lobes [32,33]. Therefore, we mimicked the efficient feature of the lawn to provide efficient and enough laser absorption in the nanostructure we created. Through previous studies, we confirmed that Ag-NG was generated in random directions and had a structure similar to grass when an overvoltage was applied using an electrochemical technique [34]. We conducted scanning electron microscope (SEM) analysis, finite-element-analysis-based electromagnetic simulation, and optimization and confirmed that Ag-NG was well fabricated and had high SERS efficiency. In addition, during the process of sample preparation, various factors, such as sonication and the presence or absence of a polydimethylsiloxane (PDMS) mold, that had a significant influence on the experimental results were investigated. We thus optimized the sensing conditions of CPF and detected it successfully in water at concentrations ranging from 100 µM to 10 nM and also at concentrations up to 10 nM. The bio-inspired Ag-NG developed in this study has the potential to be used as a practical CPF sensor in water or as a basic technique in the future.

## 2. Materials and Methods

### 2.1. Materials and Methods

All the chemical reagents (KAg(CN)_2_, bisphenol E (BPe), and CPF) were purchased from Sigma-Aldrich (St. Louis, MO, USA). Slide glass (Paul Marienfeld GmbH & Co. KG, Lauda-Königshofen, Germany) was used to fabricate the SERS substrate. All glassware was washed with piranha solution in concentrated sulfuric acid (98% *w*/*v*) and hydrogen peroxide (30% *w*/*v*) prior to use. All the solutions were prepared using Millipore deionized water with a resistivity of at least 18.2 MΩ cm at 25 °C.

### 2.2. Fabrication of Bio-Inspired Ag-NG Using Electrochemical Deposition

We synthesized Ag-NG using a previously reported synthetic method [34] wherein thin gold films were deposited using an electron beam evaporator on Si (100) wafers for electrode fabrication. The substrates were sequentially washed with acetone, ethanol, and deionized water before the Ag-NG deposition. The electrochemical deposition was performed using a conventional three-electrode system with a Pt wire counter electrode, a Ag/AgCl (1 M KCl) reference electrode, and a thin gold film-deposited substrate as the working electrode. A silver-plating solution consisting of 20 mM aqueous silver salt was prepared. Ag-NG was fabricated at −2.0 V using Ag/AgCl electrodes until the electric charge Q reached 150 C.

### 2.3. SEM and FEM-Based Electromagnetic Simulation of Bio-Inspired Ag-NG

The morphology of Ag-NG was studied using field-emission scanning electron microscopy (FE-SEM; JSM 6500-F, JEOL, Tokyo, Japan). The image was obtained using SEM under the following conditions: tilting 60° and HV 15 kV at 10,000× magnification. The interaction between the EM waves and Ag-NG was confirmed through electromagnetic simulation based on finite element analysis by using commercial COMSOL Multiphysics 5.3a. The 3-dimensional model of the Ag-NG was designed using the Rhino 5.0 program to have similar geometrical features to the Ag-NG observed in the SEM images. With the Rhino plug-in, random variables were created to create random 3D vectors. To represent the randomly directed Ag-NG as accurately as possible, the 3D vector was slightly bent in random directions. Each 3D vector constituted the skeleton of each Ag-NG. In addition, we added a shell composed of a cylinder and globular end-tip following the 3D vectors. Through these processes, we created similar geometrical features to real Ag-NG. To solve the time-harmonic Maxwell equation, we set up the x-polarized plane wave propagating along the *z*-axis, which has a 785 nm wavelength as an excitation source. As in the experiment, a 785 nm incident laser was irradiated on the substrate. In previous work, after measuring the dielectric functions of Au and Ag, the relative permeability, relative permittivity, and electrical conductivity of Au and Ag were set up [11,34]. To solve the harmonic Maxwell equation, the boundary condition under geometry was defined as global.

### 2.4. Optimization of the Sample Preparation

The degree of dispersion of CPF was considered to be a variable of the sampling condition and optimized through sonication. After the preparation of the CPF stock solution (1 mM in methanol), sonication was performed at high intensity in a water bath for 10 min to enhance the dispersion of CPF molecules in the solution. The CPF stock solution (1 mM) was diluted for further optimization and concentration experiments. In each dilution step, the sonication was conducted in the same manner.

An optimization experiment was then performed on the sampling conditions. The diameter of the manufactured circular Ag-NG substrate was approximately 4 mm. A PDMS mold was used to determine the SERS detection ability according to the volume of the CPF sample solution. The Ag-NG substrate was covered with the PDMS mold, which we created to have 18 holes (diameter: 5 mm, height: 3 mm) for the sample preparation. The Ag-NG structures were not covered due to the PDMS holes and reacted with the CPF solutions. The 40 µL of CPF solution was filled and reacted in the 18 holes in the mold. In addition, a small volume of the sample was reacted by dropping 2 μL without using the mold.

For the application of the Ag-NG sensor in the environment, we performed the test using chlorpyrifos-spiked tap water. Because the solubility of CPF molecules is very poor in water (about 3.8 µM in water), we prepared 10-fold CPF stock solutions dissolved in methanol. Then, we mixed the CPF stock solution with deionized water with a 1:10 ratio to produce the desired concentration [31]. Furthermore, we diluted the CPF solution to confirm the sensing ability of the Ag-NG sensor in tap water.

### 2.5. CPF Detection Using Bio-Inspired Ag-NG

A sample solution was prepared by dispersing the CPF powder in a methanol solution. Then, the CPF solution of each concentration that optimized the sampling condition was dropped on the Ag-NG SERS substrate and fully dried. The substrate, on which 2 µL of the sample solution was dried, was analyzed using Raman spectroscopy (InVia Reflex, Renishaw, Wotton-under-Edge, UK). All the CPF Raman spectra were measured with a 785 nm laser at 50% power via three accumulations with 1 s exposure (range: 790–1855 cm^−1^). After the analysis, we decided on the limit of detection by calculating the standard deviation of a blank control sample and the sensor response of the Ag-NG according to the CPF concentrations [35].

## 3. Results and Discussion

### 3.1. Fabrication of Bio-Inspired Ag-NG as a SERS Substrate

In this study, we fabricated a biomimetic SERS substrate for the efficient detection of CPF. In Figure 1A, the upper and lower lobes are shown as standing vertically and lying horizontally, respectively. This structural characteristic maximizes photosynthetic efficiency by allowing the incoming light to be sufficiently transmitted to the lower lobe without being blocked by the upper lobe. We attempted to replicate this principle to maximize the SERS efficiency through the Ag-NG that grew randomly in the direction of the *z*-axis. As shown in Figure 1B, we used Ag to fabricate the nanograss, which had a structure similar to that of a lawn, because Ag is the most popular noble material in the optical research field. Figure 1C,D show the images of a flat Au substrate and Ag-NG, respectively. We designed and deposited a flat Au substrate so that various experiments could be performed on each circular spot (please see the details in Section 2). Ag-NG was electrochemically deposited on the flat Au surface by using a flat Au substrate as the working electrode, a Pt wire as the counter electrode, and an Ag/AgCl electrode as the reference electrode. When the Ag-NG was well-formed on the flat Au substrate, it changed to a distinct white color that was confirmed by the naked eye.

To compare the SERS efficiency of the deposited Ag-NG, finite-element-analysis-based electromagnetic simulations were performed on the flat Au substrate and Ag-NG (Figure 1E,F and Appendix A). When a laser of 785 nm was applied, the intensity and distribution of the electromagnetic field in each substrate were confirmed (color bar: ~0–1.2 × 10^−4^ V/m). As shown in Figure 1E, no significant amplification occurred as a result of laser application to the flat Au substrate. However, as shown in Figure 1F, high SERS amplification was observed between each Ag structure and at the ends in the case of the Ag-NG. Both Ag and Au are good Raman-active materials for SERS sensors; however, the bio-inspired Ag-NG had a structure that stood vertically and had a large aspect ratio, similar to a nanorod. It is well known that Ag nanorods are good SERS-active nanomaterials owing to their structure [36,37,38]; hence, the Ag-NG showed good SERS activity in comparison with the flat Au substrate in the simulation results.

To confirm the experimental simulation results, we performed SERS analysis using a Raman indicator (1 mM of BPe) on Si glass, a flat Au surface, and bio-inspired Ag-NG. There was no significant SERS spectrum of BPe on the Si glass and flat Au surface, as is clearly shown by the magnified SERS spectra in Figure 1G,H. However, the bio-inspired Ag-NG showed greater SERS intensity than the others owing to the numerous hot spots in the substrate (Figure 1I). The SERS intensities of the main characteristic peak (~1200 cm^−1^) of BPe were compared for the different substrates and the values were 2.39 for Si glass, 138.55 for the flat Au substrate, and 18,044.96 for Ag-NG. These results are shown in Figure 1J. Although the droplets were clearly visible in the optical microscopic images, the SERS spectrum of CPF could not be obtained for the flat substrate (Appendix A) without nanostructures, such as in Ag-NG.

### 3.2. Optimization of Sample Preparation Process with Regard to Sample Dispersion Using Sonication

CPF is white, crystalline, and insoluble in water. Therefore, we dissolved it in methanol before use and it was thus necessary to optimize the process of preparation of the sample for its effective detection. The first step of optimization was sonication to obtain a sharper and clearer SERS spectrum. As CPF dissolved in methanol was used, we had to consider its dispersion state in the solution. We prepared two solutions, namely, un-sonicated and sonicated CPF, each with a concentration of 1 mM. The Raman spectrum of un-sonicated CPF (Figure 2A) did not show its characteristic peaks, including those at 1265 cm^−1^. This might have been because the CPF molecules were not well-dispersed state in the methanol solution at the molecular level, although it looked like a well-dispersed state to the naked eye. When the CPF sample solution, which was not in a well-dispersed state, was treated and dried on the Ag-NG surfaces, the CPF molecules might not have been uniformly located on the surface. For this reason, we performed sonication so that the CPF solution would have a well-dispersed state and be located uniformly on the Ag-NG surface. By contrast, the SERS spectrum of the CPF that was sonicated for 10 min showed clear and sharp main characteristic peaks (Figure 2B). For comparison, Figure 2C shows the SERS intensities of the main characteristic peak (1265 cm^−1^) for the two cases. The un-sonicated CPF sample showed a relatively low SERS intensity of 2315.28 (denoted as w/o sonic). However, the sonicated sample showed a high SERS intensity of 12,878.63 (denoted as w/o sonic). Further, we performed an experiment in which CPF was dissolved in ethanol in the same manner, and the results were identical to those of CPF dissolved in methanol (Appendix A). This indicated that sufficient sonication was important for the uniform dispersion of CPF in the solution. Thus, it was confirmed that CPF in a well-dispersed state not only facilitated sample processing but also had a higher SERS intensity.

### 3.3. Optimization of the Sample Preparation Process with Regard to Sample Drying

As the second step of optimization, we compared the following methods of sample drying: (1) PDMS mold used as a reservoir and (2) the direct dropping method. First, to obtain a comparison control solution, we obtained the Raman spectrum of CPF powder that was not dissolved in methanol (Appendix A). The spectrum showed characteristic peaks similar to those that were previously reported [29,31]. The results in the case of using the PDMS mold in the sample preparation process are shown in Figure 3A,B. PDMS molds are widely used in sample preparation because they are sticky, hardly deformed, and chemically stable. We used PDMS molds on the surface to dry CPF for 6 h in a desiccator at room temperature. A total of 40 µL of CPF solution was dried in PDMS molds on Ag-NG. Similarly, the control solution was also dried (inset of Figure 3A). No other peaks related to the characteristic peaks of CPF were observed in the case of the control solution, confirming that the substrate was in a clean state (Figure 3A). The results for the control solution showed only the methanol Raman spectrum, along with some experimental errors. The Raman spectrum of CPF is shown in Figure 3B, and it can be seen from the figure that there were no other distinguishable main characteristic peaks, including the peak at 1265 cm^−1^. Figure 3C,D show the results of the direct dropping method without the PDMS mold. This method had the advantage that the sample dried quickly in a few minutes owing to the very small amount (2 µL) and absence of the mold. However, there was a disadvantage in that it was difficult to concentrate the sample on the substrate and detect very tiny quantities of it when the sensitivity of the sensor was poor. Figure 3C shows the Raman spectrum of the control solution when the direct dropping method was used. It is similar to that shown in Figure 3A as the sample solution was treated directly on the SERS substrate, and thus, no influence of methanol was observed; however, as shown in Figure 3D, we obtained a clear SERS spectrum of the main characteristic peaks of CPF, such as 1140, 1265, and 1448 cm^−1^. To compare the influence of the presence of PDMS molds in the sample-drying process, we compared the SERS intensities of the main characteristic peaks of CPF (equal to 1 µM) in Figure 3B,D (Figure 3E). It can be seen in Figure 3E that among the various peaks, the SERS intensity of CPF at 1265 cm^−1^ was remarkably different. The results thus indicated that the direct-dropping method was suitable for obtaining clear and sharp peaks of CPF, even though the sample amount was very low (2 µL), owing to the superior SERS efficiency of Ag-NG. However, the sample dried by the PDMS mold method had various variables in the experiment; hence, a clear Raman spectrum could not be obtained.

### 3.4. Detection of CPF According to the Concentration Using the Ag-NG

Using the results of optimization, we prepared sonicated CPF solution and directly dropped 2 µL of it on the Ag-NG substrate without a PDMS mold for the sensitive detection of CPF according to the concentration (Figure 4). The Ag-NG was treated with CPF from 100 µM to 10 nM, including the control solution, in the same manner. Figure 4A shows the average SERS spectra of CPF at 100 µM, 1 µM, and 10 nM, as well as the control solution. Among various main characteristic peaks, we chose the 1265 cm^−1^ peak due to it displaying the strongest intensity. For a precise analysis, a *t*-test and linear regression fitting were conducted (Figure 4B and Appendix A, respectively). At first, the *t*-test results in Figure 4B shows that the SERS intensity of 10 nM of CPF has an obvious difference in comparison to that of control. As the *p*-value was 0.0001 (***), it was statistically significant. In addition, the linear regression line fitting was performed to calculate the limit of detection (Appendix A). As a result, the fitting line was as follows:Y = 0.07041 ∗ X + 387 (R^2^: 0.9919)

From the results, the limit of detection (LOD) of Ag-NG toward the CPF was determined as ~500 nM by the response to the CPF concentrations. The LOD was 500 nM, but it was sufficiently possible to discriminate 10 nM of CPF in comparison to the control via a SERS measurement using bio-inspired Ag-NG. In comparison with other sensing techniques, the Ag-NG SERS sensor showed similar sensitivity despite the non-usage of chemical reagents (Appendix A). The results indicated that our method could detect CPF directly and sensitively without chemical target probes, such as acetylcholinesterase, and did not involve a complex process of sample preparation. In addition, remarkably, we succeeded in the detection of CPF molecules in tap water (Appendix A). Considering the SERS intensities of them, it might have enough potential to detect the very low concentrations of CPF in the future.

## 4. Conclusions

In summary, we performed the high-sensitivity detection of CPF using Ag-NG by mimicking the structure of a lawn that exhibited efficient photosynthesis owing to the structural characteristics of its upper and lower lobes and had a large surface. We mimicked these features of the lawn and developed Ag-NG that allowed for direct detection of chlorpyrifos without chemical reagents with very high SERS efficiency. To confirm this, we performed SEM analysis, FEM-based electromagnetic simulation, sample optimization, and considered the dispersion effect via sonication. The results confirmed that when a PDMS mold was used during sample preparation, the integration rate of CPF on the Ag-NG substrate was significantly lowered, which was disadvantageous for detection. In addition, it was confirmed that the dispersion of CPF through sonication increased the efficiency of SERS detection. Through the optimization process, the high-sensitivity detection of CPF was carried out according to its concentration and up to ~500 nM was successfully detected. Additionally, we could sufficiently discriminate 10 nM of CPF compared to the control. Through the high sensitivity of the bio-inspired Ag-NG, remarkably, the CPF in the tap water was detected successfully. Despite silver displaying a strong SERS intensity, because of its weakness toward oxidization, it still has limitations regarding the aspect of stability. Therefore, there is the possibility to improve our sensor in future work by enhancing the stability of the sensor. In our opinion, the results of this study have the potential to be used in various environment toxicants sensors based on biomimicry and SERS spectroscopy and biomimicry.

## Figures and Tables

**Figure 1 materials-15-03454-f001:**
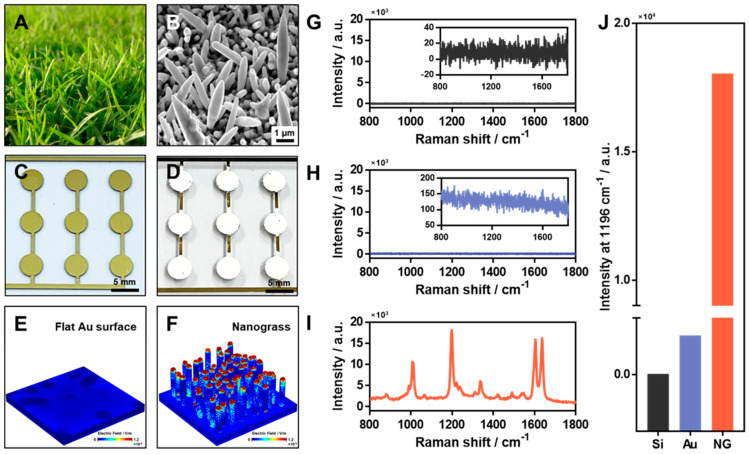
Bio-inspired Ag-NG as a sensitive surface-enhanced Raman spectroscopy (SERS) substrate. Optical and SEM images of (**A**) a lawn and (**B**) Ag-NG (scale bar: 1 µm). Optical images of a (**C**) flat Au substrate and (**D**) Ag-NG fabricated substrate (scale: bar: 5 mm). FEM-based electromagnetic simulation results for (**E**) a flat Au surface and (**F**) Ag-NG (color bar: ~0–1.2 × 10^−4^ V/m). SERS spectra of bisphenol E (BPe, 1 mM) for different substrates: (**G**) Si glass, (**H**) flat Au surface, and (**I**) Ag-NG. The insets in (**G**,**H**) are magnified spectra of BPe for the respective substrates. (**J**) Comparison of SERS intensity of a BPe main peak (1196 cm^−1^) for different substrates (Si: slide glass, Au: flat gold substrate, and NG: Ag-NG).

**Figure 2 materials-15-03454-f002:**
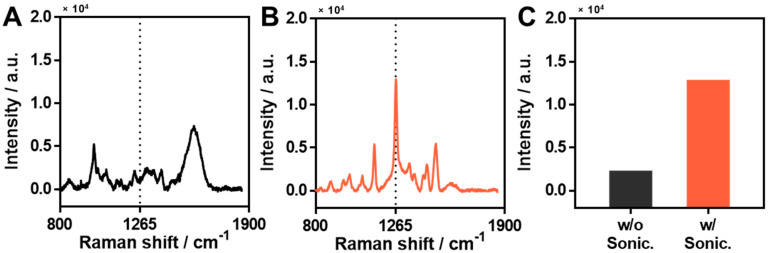
Optimization of sample preparation: sonication. (**A**) SERS spectrum of un-sonicated CPF solution on the Ag-NG. (**B**) SERS spectrum of sonicated CPF solution on the Ag-NG. (**C**) Comparison of SERS intensity at 1265 cm^−1^ according to sonication. CPF was dissolved in methanol and the solution was sonicated for 30 s. The sonicated and un-sonicated CPF solutions (1 mM) were treated equally on the Ag-NG.

**Figure 3 materials-15-03454-f003:**
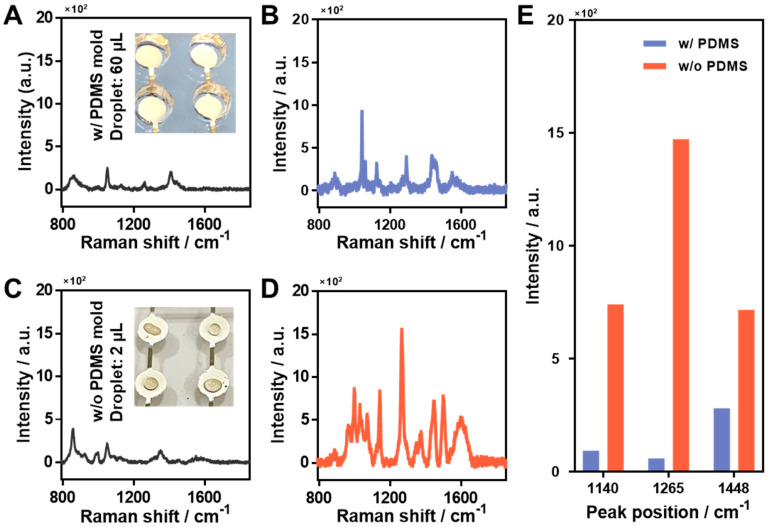
Optimization of sample preparation: sample treatment. The case of using a PDMS mold in the preparation process of (**A**) control solution and (**B**) CPF 1 µM treated Ag-NG substrates. Without a PDMS mold: (**C**) control solution and (**D**) CPF 1 µM treated Ag-NG using the direct-dropping method (single droplet: 2 µL). The inset images are the optical images of each sample with and without a PDMS mold. (**E**) Comparison of the main characteristic peaks (1140, 1265, and 1448 cm^−1^) of CPF according to the sample preparation condition with and without a PDMS mold.

**Figure 4 materials-15-03454-f004:**
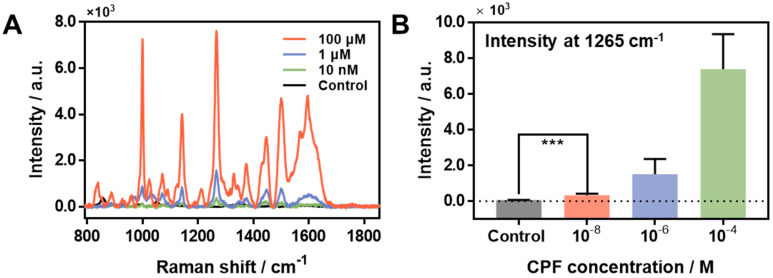
SERS measurement of CPF for various concentrations from 100 µM to 0 M. (**A**) SERS spectra of CPF according to the concentration. (**B**) SERS intensities of the main characteristic peak (1265 cm^−1^) of CPF depending on the concentration from 100 µM to control. (Limit of detection: 500 nM, t-test result: *** *p*-value = 0.0001.)

## Data Availability

Not applicable.

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
