# Peer review of "Detection of Chlorpyrifos Using Bio-Inspired Silver Nanograss"

_materials, 2022, doi:10.3390/ma15103454_

Round 1
Reviewer 1 Report
The authors wrote a paper: "Highly Sensitive Detection of Chlorpyrifos using Bio-Inspired Silver Nanograss".
The main goal of this manuscript is to detect Chlorpyrifos using surface-enhanced Raman spectroscopy (SERS) with rapid fabrication of biomimetic Ag nanograss as a substrate.
Introduction is extensively written and covered with many references. In my opinion, there is no need for changes.
Page 3.
2. Materials and Methods
This sentence should be rewritten: "The geometry of the Ag nanograss was implemented through the 3D CAD program Rhino 5.0 to confirm its structure through SEM to be as similar as possible."
Page 3.
2. Materials and Methods
What does it mean: "by filling the mold so that the sample was full with 40 μL"?
Page 4.
3. Results and Discussion
Figure 1.
What are the units of the Raman intensities? Why did the authors put "a.u."?
And in spectroscopy there is a convention red-to-the-right, so the spectra should be inverted.
Abbreviation NG is not defined.
Page 4.
3. Results and Discussion
Figure 1.
How were the intensities on J determined?
Page 8.
Figure 4.
Linear fitting presented on Fig. 4 are meaningless, it is clear that intensities do not follow general linear model. Blue dots are even not visible on the figure. How exactly did the authors determine LOD? Detailed explanation should be provided in 2. Materials and Methods. LOD abbreviation is never defined in the paper.
Page 11. Conclusion
Authors claim: "This study may prove to be a milestone in the field of sensor research based on biomimicry and will pro-vide the groundwork for research on environmental pollutant detection."
This is highly speculative interpretation of the presented work and should be excluded from the paper.
Author Response
Thank you for the valuable reviewer's comments. Please see the attachment for the response to your comments.

Reviewer 2 Report
The manuscript entitled "Highly Sensitive Detection of Chlorpyrifos using Bio-Inspired Silver Nanograss". I do see good importance in this work; the topic is relevant for both materials and environmental fields. However, compared to the previous reports published in the literature, this work lacks some important information and discussion. Before I recommend its acceptance, some points must be clarified and a moderate revision is needed.
Some other issues that need to be addressed are:
- Please explain the problem that you want to solve and the contributions of the study in the abstract.
- The main problem statement and justification for the research have not been clearly stated.
- Is not clear the contribution of the manuscript to the empirical literature.
- Would you explicitly specify the novelty of your work? The main novelty in this work must be clearly pointed out in the introduction.
- The authors should mention the concept of this work with the progress against the most recent state-of-the-art similar studies.
- The limitation of this study needs to be provided as well.
- “Figure 1a shows that a lawn not only has a large surface per unit area…” what does that mean? SEM does not provide any clue on surface area….per unit? This is a wrong statement. Correct the sentence.
- It lacks comparison with the literature. A table would be welcomed. Comparing the main outcomes of this work with some of the literature.
- All conclusions must be convincing statements on what was found to be novel, and impactful based on the strong support of the data/results/discussion.
Author Response

(The authors gave the same response as above.)

Reviewer 3 Report
Major comments:
- There is no real sample detection analysis, but only calibration (Fig.4, part 3.4). The detection of Chlorpyrifos should be demonstrated.
- There is no comparison of of Chlorpyrifos detection by other (bio)analytical systems. What did SERS implementation improve (sensitivity, simplicity or duration of analysis)?
Minor comments:
- The first paragraph of Introduction (p.1-2) should be rewritten to be more concise. Only relevant information should be remained. Overall the Introduction section could be modified for clearness and consistency.
- The statement that Chlorpyrifos acts by “inhibiting the production of acetylcholinesterase” (p.2) is totally wrong. Not enzyme production, but inhibition of enzymatic activity happens. The text should be corrected.
- Expenses of enzymes and DNA (p.2) is a weak discussion as compared to gold and silver used in the work. It can be omitted without any disruption in consistency.
- Information given in Section 2.3 (p.3) is insufficient for reproducibility (of work). It should be presented with more details.
- First mention of “CPF stock solution (100 μM)” (p.3) made me wonder what solvent was used. It become clearer several paragraphs later. I think it will be better to specify the solvent in the first place.
- “This is because although CPF powder appeared to be well dissolved in methanol when seen with the naked eye, it might not be so.” (p.5): This is quite strange at least. The solution was dried before analysis if I understand correctly (last paragraph on p.3). So, the authors analyzed samples of solid films anyway (with reliable SERS signal) and this has no logical connection with solubility or “naked eye” detection on CPF itself. It seems something has happened with CPF during sonication. It is necessary to confirm the presence of (unmodified) analyte after sonication in the first place while, for example, analyzing it by HPLC/MS, GC/MS or any other convenient method.
- The statement “The spectrum showed characteristic peaks similar to those that were previously reported [25,29,31].” (p.6) is wrong. Ref.25 has used another analytical range and thus there are no peaks which were utilized in the current work. In the Ref.29 only two peaks 1110 and 1270 cm-1 are comparable. In the Ref.31 similar peaks were 1150 and 1440 cm-1. Although both Refs.29 and 31 utilized another wavelengths for analysis.
- PDMS exercises are looks like that PDMS absorbs an analyte from the substrate. That’s all. If it is true I don’t understand why it is so necessary and/or present in the main text.
- On Fig.4a (p.8) there is no value of factor (like 100, 1000, etc.). The correlation of Intensity vs. Concentration (Fig.4b) is highly nonlinear, even in semilogarithm coordinates. To approximate it by linear function is very bad idea. Correlation coefficients (which are absent for Eq.1-3) are poor and such equations are unreliable/unusable. Also, there is no error (or standard deviation) bars on Fig.4b which should be added. Authors should use another (proper) function for approximation.
- The statement “detection limit was 10 nM” (p.8) is highly questionable. Authors should look for IUPAC determination of “detection limit” and recalculate it accordingly. Also, authors should specify which wavelength was used for that analytical detection limit.
- I don’t understand after reading the manuscript what “efficient photosynthesis” (p.8 and elsewhere) has something to do with the subject of the work. Also, sample was not optimized (instead of the statement, p.8), and I suggest to check the text again (and English spelling also) and correct such lapses.
- Something odd has happened with Refs. 31, 32 and 36.

Author Response

(The authors gave the same response as above.)

Round 2
Reviewer 1 Report
No additional comments.
Author Response
The quality of our discussion has improved thanks to the reviewer's valuable comments.
Reviewer 2 Report
The manuscript was revised accordingly and improved. I recommend the publication of the manuscript in its present form.
Author Response

(The authors gave the same response as above.)

Reviewer 3 Report
Text on P.2 (lines 45-54); P.2 (lines 82-89); and P.4 (lines 162-169) is a self-repeat. I suggest authors to modify the text.
P.3, lines 120-132: Details on computational method are still insufficient.
Figure 2 and Figure 4: Why does peak at 1265 cm-1 corresponding to 100 μM have intensity ca. 30,000 on Fig2B and ca. 13,000 on Fig2C and ca. 7,700 on Fig4A and ca. 1,800 on Fig4B (100 μM=10-4M)? What is it? How it is possible to talk about some calibration curve and detection after such jumping in values of same concentration?
P.8, lines 308-309, “In addition, remarkably, we succeeded the detection of CPF molecules in the tap water (Figure. S5)”:
Firstly, I wonder how was it possible to dissolve millimolar concentrations (1 mM and 5 mM) of chlorpyrifos in tap water? Its solubility in water is in micromolar range.
Secondly, when analyzing some approximation to a ‘real’ sample, the authors have obtained a decreased signal (at least, 5-fold). That is what authors should highlight and catch the Reader’s eye. This additionally argues to obtain ‘calibration curve’ under conditions close to the real ones.
Taking into account the Response 9 (“Thank you for your valuable comment. As you mentioned, it is not correct general linear fitting because the ‘x-axis’ of Fig. 4b has the log scale according to the concentration of chlorpyrifos. Indeed, the results about the SERS intensities according to concentration were in the graph below which has a linear X-axis (Fig. R2, reviewer only). However, it still be meaningless, because it does not have the standard deviations for calculating the limit of detection. Because the spectra and SERS intensities were measured just one according to the concentrations. So, following your comment, we deleted the linear fitting curves and equations in Results and discussions and Fig. 4, respectively.”) and Response 10 (“Thank you for your valuable comment. We decided the limit of detection qualitatively, which means the minimum concentration of CPF measured by our technique. In detail, the 'qualitative detection limit' is the minimum concentration of CPF which can be analyzed with a clear difference in SERS intensity from that of the control. It was not correct to use “detection limit” in the manuscript instead of the qualitative detection limit. Strictly, the "Detection limit" which was determined in IUPAC was not suitable for our results. Because the spectra and SERS intensities were measured just one according to the concentrations. So, we can't calculate the "Detection limit". Instead of the calculated detection limit, we used a qualitative detection limit. The qualitative detection limit was equally reasonable as in the example in the reference to the SERS sensor [Lu, Han, et al. "Ag nano-assemblies on Si surface via CTAB-assisted galvanic reaction for sensitive and reliable surface-enhanced Raman scattering detection." Sensors and Actuators B: Chemical 304 (2020): 127224.]. In addition, we added the information about the decision on the qualitative detection limit of our sensor in the Material & Methods section.”) of the authors, I need to note that both responses are highly interconnected and concern a single problem, namely, calculation of “detection limit”. I will begin from the end. Firstly, there is no reason to invent and implement new and ambiguous terms (like “qualitative detection limit”). There is term “quantification limit” which authors desired to re-invent unconsciously. However, it is actually determined in a similar experimental way as “detection limit”. Secondly, both terms were implemented in spectroscopy a long-long time ago. Newly-fashioned SERS is only one of its sort. Thirdly, to reliably compare results of many works (or various analytical systems), the authors should use the same ‘characteristic’, determined by the same way.
Thus, the authors should not apply the term “detection limit” at all. Alternatively, the authors should determine its value properly.
I see there are difficulties with it (especially with determination of standard deviations), so I will try to help/explain. For example, the authors can have 100 spectral scans of methanol alone (i.e. ‘blank’) and enumerate 100 intensity values at certain wavenumber. Then, the authors can treat this sampling group statistically (manually or with software) and get both the ‘mean’ and the ‘standard deviation’ of the blank measures. For usual significance level (0.05) and large enough sampling (normal distribution), the coefficient will be close to 3 (i.e. ‘3-sigma rule’). Then, authors should similarly treat their sample group (i.e. “10 nM CPF”) and calculate the mean value of intensity. If it will be less than ‘mean blank’+3*SD, then “10 nM CPF” is NOT a detection limit and vice versa. For a quantification limit, similar coefficient is usually equated to 10.
Figure 4: The authors discussed detection limit of 10nM=10-8 M. Please, look at Fig.4B. There is absolute ZERO at 10-8M and absence of data at all. Interesting, that detection in real samples were made by the authors with 1 mM (10-3M) and 5 mM (5x10-3M), and in accordance with Fig.4B they should see the intensity ca. 8,000, but in accordance with Fig4A same intensity corresponds to concentration 100 μM= 10-4M. So, concentrations differed in 10 times can give same intensity. That is bad result for the detection method.
Author Response

(The authors gave the same response as above.)

Round 3
Reviewer 3 Report
I highly appreciate the efforts of the authors of this article to improve the quality of the scientific material presented by them, which led to the fact that the illustrative material and the text of the article have changed significantly. Instead of 10 nM, the limit of determination is now 500 nM, that is, it has increased 50 times. In this regard, I propose to modify the title of the article by removing the words "Highly Sensitive" from it and leaving this option: "Detection of Chlorpyrifos using Bio-Inspired Silver Nanograss". I think that this title will be enough to attract readers' attention to this article, but at the same time the title of the article will be correct and better reflect the data obtained, which now do not exceed previously known results in detection of Chlorpyrifos, but are comparable with them, according to Table S1 and the new text of the authors (Lines 310-311). The same phrase " highly sensitive " should be removed from the sentence "…we try to develop a highly sensitive SERS platform that can detect the chlorpyrifos directly…" (Line 16, Abstract). In the rest of the text, this phrase has already been correctly deleted by the authors. This is a minor remark that should be taken into account, and further I recommend accepting the article for publication.
Author Response
The quality of our discussion has improved thanks to the reviewer's highly valuable comments.